# OpenReview forum: "Measuring And Improving Persuasiveness Of Large Language Models"
_ICLR.cc/2025/Conference — ICLR 2025 Poster_

### Official Review · Reviewer_Xibh · 2024-11-02

**Soundness:** 3
**Presentation:** 3
**Contribution:** 2
**Rating:** 8
**Confidence:** 4

**Summary:**

This paper tackles the challenge of evaluating, generating, and transforming persuasive messages. Based on a large volume of tweets, the authors constructed pairs of messages—one more persuasive than the other—to create the PersuasionBench benchmark. Additionally, the paper introduces PersuasionArena, an evaluation platform to measure the effectiveness of persuasive messaging. Together, these resources provide a foundation for systematically assessing and enhancing language models’ capabilities in generating and adapting persuasive content.

**Strengths:**

- The method for collecting pairwise data is both interesting and scalable. The authors apply a rigorous filtering process, resulting in a dataset with substantial potential for future research in persuasive language modeling.
- The paper presents a diverse range of experimental tasks, including various forms of transsuasion and both generative and simulative settings. This breadth in task design highlights the robustness of their approach to generating and evaluating persuasive messages.
- The domain transfer experiment is particularly compelling. Through a live experiment conducted via an app, the authors demonstrate the practical impact of the proposed systems and emphasize the model's real-world applicability in dynamic, user-driven contexts.

**Weaknesses:**

Below is my original review. The authors have addressed these issues, so I've updated my score.

The truthfulness of transsuaded tweets is still somewhat unclear. As an attempt to make a tweet more persuasive, the model sometimes seems to add new content and statements (including statistics). It would be nice to discuss the degree of truthfulness of these statements, its potential impact, and some mitigating methods.

===

- In the transsuasion task, it is unclear if the semantic meaning is consistently preserved between paired messages. For example, the two tweets in Figure 1 seem to diverge in content, which raises questions about their semantic similarity.
- Some important evaluation seems missing. The paper does not sufficiently address whether semantic meaning is retained in generated messages for transsuasion. Additionally, for the generation tasks, it remains unclear if the generated outputs consistently adhere to the specified conditions.
- The paper provides minimal discussion on the experiment results, providing not enough scientific insights.
- It is not discussed the characteristics of model-generated messages and the factors that improve their persuasiveness. Are there be any simple factors like the length of a message and the existence of hashtags or images? A deeper analysis into which domains the model excels in and where it falls short would enhance understanding of its strengths and limitations.
- Although the paper covers a wide range of settings, the presentation is somewhat unclear. The extensive descriptions make it challenging to retain all details, and a clearer, more structured organization could improve readability.
- The authors do not mention whether the collected dataset or source code will be publicly available, which could limit reproducibility and further research.
- While this focus is not inherently a drawback, a clearer explanation of this choice and its implications would be beneficial.
- Some typos and formatting issues: line 087 (citation format), line 299 (”any model but can be conclusively”), line 398 (citation format), line 421 (citation format).

**Questions:**

- Some settings require adding images. How did you obtain or generate those images?
- Line 349: What is used for the oracle in the “Oracle as a judge” setting?
- Table 2: What does “ep” stand for?
- Table 2: What is the Oracle setting here?
- Table 2: What is the difference between “Ours Instruct” and “Ours (CS+BS+TS)”? That is, what is the difference between the instruction tuning setting and non-instruction tuning settings?
- Table 3: What does “it” stand for?

---

> ### Author Response · Authors · 2024-11-21
> **Thank you for your thoughtful review**
>
> Thank you for your thoughtful review and suggestions on how we can improve the paper. They are very helpful for improving the overall work.
>
>
> >  if the semantic meaning is consistently preserved between paired messages. for the generation tasks, it remains unclear if the generated outputs consistently adhere to the specified conditions.
>
> The question about transsuaded text having similar meaning pertains to the degree to which AI can operate on messages without the sender’s supervision. This could be considered as “autonomy” of AI agent ([3] Hancock et al). We define nine types of transsuasion (Table-1) varying the autonomy of the AI agent. E.g. autonomy can be restricted to changing just the image (VisOnly), or just paraphrasing text (Parap).
>
> In the table below, we show the similarity metrics between transsuaded tweets obtained or generated from (1) ground truth (GT) (2)  GPT-4o (3) Ours(13B) (4) Vicuna(13B). We use the following metrics to test semantic similarity between transsuasion pairs of the ground truth and generated samples:
>
> 1. NER Match: The number of NERs that are same on both sides divided by total NERs
> 2. Factuality match: We prompt GPT-4o to assess if both tweets do not differ factually, we also ask it to give a confidence rating on a scale of 5 tell if it is sure about both the facts being different and only consider the pairs where the confidence is at least 4.
> 3. MetricsMatch: The proportion of pairs of tweets that follow the constraints we created in Table 1
>
> |Model\Metrics|**NER Match**|**Factuality Match**|**MetricsMatch**|
> |--|--|---|--|
> | **GPT-4o** | 97.8% | 94.1% | 87.6% |
> | **Vicuna (13B)** | 92.7% | 84.2% | 80.1% |
> | **Ours (13B)** | 92.1% | 93.6% | 85.2% |
> | **Ours (DPO) (13B)** | 94.9% | 94.3% | 87.2% |
> | **GT** | 87.1% | 88.3% | - |
>
> The tables show that not only are the input pairs largely similar semantically, but also that the output pairs are more controlled. Thanks for the suggestion we have added this as table 15 in the appendix.
>
>
> > Win rate across topics, A deeper analysis into which domains the model excels in and where it falls short
>
> We extracted topics from the username and Twitter bio using BERTopic. These topics were clustered and assigned a name by GPT-4o-mini and visualized using a sunburst graph (Figure 5 in the new version of the paper). Similarly, we also analyzed the topic distribution of tweets (Figure 6 in the new version of the paper).
>
> **Win rate across topics:** In Figure 8, we present our highest and lowest average win rates against GPT-3.5, GPT-4, and GPT-4o across various topics. Our performance reaches its peak, with win rates close to 80%, on topics such as retail product innovation, e-commerce sales, and new launches. Conversely, on topics where we have the lowest win rates—such as market analysis, platform growth, and fashion in tech—GPT-4o performs comparably or slightly better.
>
>
> > characteristics of model-generated messages and the factors that improve their persuasiveness. Are there be any simple factors like the length of a message and the existence of hashtags or images
>
> What makes a tweet more persuasive? We observe that the correlation between tweet length and persuasiveness is negligible (0.04, p-value = 4.88e-65), and linguistic features such as emojis, sentiment, and hashtags have minimal influence (see Figure 7 in the new version).
> However, we do observe certain brand specific insights in model generated tweets. To extract these brand specific insights we first cluster the embedded (using RoBERTa) tweets of a brand and summarize the insights from a cluster using gpt-4o-mini. We have added them below (also in appendix section B.5)
>
> - Bulgari
>   - Transsuaded tweets evoke strong emotional engagement and vivid imagery
>     - Example: https://x.com/Bulgariofficial/status/1856736301657235947
>   - Transsuaded tweets emphasize products rather than events.
>     - Example: https://x.com/Bulgariofficial/status/1843573678736584907
>   - Transsuaded tweets showcase a unique and innovative design element.
>     - Example: https://x.com/Bulgariofficial/status/1846936730471129102
>
> Similarly, we have extracted rules for some more brands and industries below
> - Starbucks
>   - Transsuaded tweets emphasize a seasonal theme or promotion.
>   - Transsuaded tweets convey a personal experience or sentiment.
> - Nike
>   - Transsuaded tweets indicate urgency or time sensitivity
>   - Transsuaded tweets include a specific date and time.
>   - Transsuaded tweets emphasize a specific cultural or historical significance.
> - Airbnb
>   - Transsuaded tweets evoke a nature-centric experience.
>   - Transsuaded tweets emphasize a specific location or city.
>   - Transsuaded tweets emphasize local economic impact.

---

> > ### Author Response · Authors · 2024-11-21
> >
> > > Some settings require adding images. How did you obtain or generate those images?
> >
> > For the tweets containing images, we extracted the caption for all images with LLaVA. An LLM under test in PersuasionBench or PersuasionArena takes the caption of low-performing tweet (T1’s) image as input and then generates the caption for the transsuaded tweet. We then use these captions as prompts to a text-to-image generator model (Adobe Firefly). An example of the instruction format is in Listing 12, ADDIMG task (lines 1916-1920 of the current version) and an illustration of the images generated using transsuaded captions is shown in Fig-2.
> >
> >
> > > Line 349: What is used for the oracle in the “Oracle as a judge” setting? What is the Oracle setting here?
> >
> > The Oracle is a Vicuna-1.5-13B (same as the base LLM for Transsuasion) finetuned on Train+Test data, the details are outlined in lines [342-353] of the current version, We will fix the typo in line [346] of the current version. The oracle is trained on the train and test data with the best training regime (for the comparative transsuasion tasks) to measure the persuasiveness of the transsuaded tweet.
> >
> > > Table 2: What does “ep” stand for?
> >
> > “ep” Stands for the number of epochs the model was trained for. We have clarified this in Table 3
> >
> > > Table 2: What is the difference between “Ours Instruct” and “Ours (CS+BS+TS)”? That is, what is the difference between the instruction tuning setting and non-instruction tuning settings?
> >
> > The Ours-Instruct model is trained on (BS+CS+TS) data appended with 30k explanations generated by Vicuna-1.5-13B LLM. We prompted (refer to Listing 23 for exact prompt) Vicuna-1.5-13B to generate explanations for differences between tweet T2 (high likes) and T1 (low likes) within a given pair (T1, T2). The prompt instructs the model to explain potential reasons for T2’s superior performance compared to T1. Details of the instruction task setup are provided in lines [479–485] of the current version. This task significantly enhances performance on downstream tasks.
> >
> > > Table 3: What does “it” stand for?
> >
> > “it” stands for iterations. Let's say the input tweet to the transsuasion model is T, the transuaded tweet is T_1, then T_1 is again used as an input to generate T_2 or 2 iterations and so on. Therefore 2-it refers to T_2. We have clarified this in Table 3.
> >
> >
> > > collected dataset or source code will be publicly available
> >
> > The collected datasets, source-codes, and trained models will be open sourced. Our aim is to build PersuasionArena and PersuasionBench, inspired by LMSys. LMSys is supported through public donations (https://lmsys.org/donations/) from organizations such as Hugging Face, MBZUAI, Kaggle, NVIDIA, and others. Similarly, a few organizations have already reached out privately, offering resources to support the development of open-source infrastructure for PersuasionBenchand PersuasionArena. To benchmark their models on LMSys, model developers submit either model parameters or APIs. Following a similar approach, we plan to run benchmarks and arenas where model developers can continuously evaluate their models. These resources will be released upon acceptance.
> >
> >
> >
> >
> > > Some typos and formatting issues: line 087 (citation format), line 299 (”any model but can be conclusively”), line 398 (citation format), line 421 (citation format).
> >
> > Thanks for pointing out we have made the changes.

---

> ### Author Response · Authors · 2024-11-21
>
> >  discussion on the experiment results
>
> We acknowledge the need to provide deeper insights into the experimental results. Below, we address specific aspects of our work. We have also added these to the paper (Appendix Section-F, we will move it to the main paper in the camera-ready).
>
> **Scaling Trends for LLM Persuasion:** Several studies [1,2] highlight an increasing correlation between model scale and persuasiveness. However, our fine-tuned 13B models (CS+BS+TS) with an ELO of 1304 outperforms significantly larger models on generative transsuasion, such as LLaMA-3-70B (1187) and GPT-4 (1213), GPT-4o (1251). Additionally, our 7B model (1099) surpasses GPT-3.5 (1092) in performance (refer to table 3 for detailed evaluations). These results indicate that persuasive capability is not solely determined by model scale. Instead, targeted training of smaller language models, when combined with scaling, can yield competitive or superior outcomes. From Table-3, we also see that GPT-4o beats GPT-4 by 38 elo  and GPT-3.5 160 elo points on PersuasionArena, which correspond to a win rates 55.45% and 71% according to Bayes-Elo Calculation. This is in line with the OpenAI GPT-o1 model card [2], where the win rate of the latter is 78.1% on the ChangeMyView benchmark (compared to 71% on our arena).
>
> **Training regimes for transsuasion models:** We ablate our experiments across various training regimes (finetuning (IFT), DPO), task combinations (BS+CS, BS+CS+TS), and the inclusion of self-generated explanations (Ours-Instruct). Our findings reveal that multi-task IFT (BS+CS+TS) achieves the best ELO (1304) on generative transsuasion. In contrast, DPO trained on TS samples performs slightly lower (1283) overall but demonstrates a marginal advantage of in similarity on NER Match(+2.8%), MetricsMatch (+2.5%), and FactualityMatch(+0.7%)  (refer to Table 15). Training exclusively on BS and CS yields the highest performance for behavior simulation (62.2%). While the addition of self-generated explanations does not significantly affect BS, CS, or TS individually, it notably enhances performance on downstream tasks including Humans as judges of persuasion(+5.5%), marketing blogs simulation (+6%), and Audience specific transcreation (+3%) (refer to Tables 7, 9, and 10 respectively).
>
>
> **Transfer of Transsuasion to Other Tasks:** To check the transfer of persuasion capabilities measured over Twitter to other domains, we test all models on 4 benchmarks: Humans as Judge (our study), Humans as Judge (Anthropic Persuasion study), Marketing blogs dwell time and views prediction, Audience specific transcreation. This also allows us to test the transfer power of Twitter-finetuned 13B model whose persuasion capabilities were developed over Twitter to other channels and domains.
>
> 1. Marketing blogs dwell time and views prediction: We improve 19% and 22% compared to base model on dwell time and views prediction respectively (refer to table 9)
> 1. Audience specific transcreation: We improve the performance ~2x compared to the base model on targeting (refer to Table 10)
> 1. Humans as judges (our study): we improve by 15%, 20% and 50% on on Upvote/Downvote classification, Reasoning classification, and Feedback perplexity compared to the base model (refer to table 7)
> 1. Humans as judge (Anthropic study): Our rank correlation is 0.47, ~6.5x more compared to the base model (0.07) (refer to table 8)
> These show that models trained on the task of transsuasion also transfer to completely unseen domains, channels, behaviors, and tasks. These findings demonstrate the robustness of our framework and the broader applicability of PersuasionArena and Bench across varied tasks involving persuasion.
>
> Further to measure the robustness of PersuasionArena we collect the rankings from all metrics (8) of the (4) transfer benchmarks above, next we calculate the mean kendall tau coefficient between PersuasionArena these 8 rankings, this gives us a mean Kendall Tau coefficient of 0.69 which is considered strong. The consistent rankings of PersuasionArena and unseen domains further validate its importance.
>
> **What makes content more persuasive:** We observe that the correlation between tweet length and persuasiveness is negligible (0.04, p-value = 4.88e-65), and linguistic features such as emojis, sentiment, and hashtags have minimal influence (see Figure 7 in the new version). However, we do observe certain brand specific insights in model generated tweets. To extract these brand specific insights we first cluster the embedded (using RoBERTa) tweets of a brand and summarize the insights from a cluster using gpt-4o-mini. We have added them in appendix section B.5

---

> ### Author Response · Authors · 2024-11-22
>
> [1] Durmus, E., Lovitt, L., Tamkin, A., Ritchie, S., Clark, J., & Ganguli, D. (2024, April 9). Measuring the Persuasiveness of Language Models. https://www.anthropic.com/news/measuring-model-persuasiveness.
>
> [2] Gpt-o1 system card." https://openai.com/index/openai-o1-system-card/
>
> [3] Hancock et al, AI-Mediated Communication, JCMC, 2019

---

> > ### Comment · Reviewer_Xibh · 2024-11-23
> >
> > Thank you for the extensive rebuttal. I think it addresses my conerns and suggestions sufficiently and I am almost willing to raise my score. But before that, I have a few more clarifying questions.
> >
> > >if the semantic meaning is consistently preserved between paired messages. for the generation tasks, it remains unclear if the generated outputs consistently adhere to the specified conditions.
> >
> > Thank you for the table and the metrics. While the numbers appear to be high, I am not very clear what is an acceptable number to judge that two tweets have the same meaning and the specified conditions are met. Can you provide some guidance? For example, what are the measurements for the first tweet in Figure 1?
> >
> > >Win rate across topics
> >
> > In Figure 8 in the revision, what model setting does "our performance" refer to? Please clarify in the paper.
> >
> > >characteristics of model-generated messages and the factors that improve their persuasiveness. Are there be any simple factors like the length of a message and the existence of hashtags or images
> >
> > I found this analysis very fascinating.
> >
> > For your brand-specific analysis for Bulari, you mention that transsuaded tweets evoke strong emotional engagement and vivid imagery, emphasize products rather than events, and showcase a unique and innovative design element. Is this conclusion based on a systematic analysis or based on a few examples? The second aspect (i.e., emphasize products rather than events), in particular, seems to somewhat contradict your observation from Starbucks (i.e., emphasize a seasonal theme or promotion). I would expect a more systematic analysis because I believe that can be an important scientific contribution of this work.
> >
> > In addition, if a transsuaded tweet becomes more persuasive because it emphasizes a seasonal theme or promotion, does it mean that the original tweet had a different message without this promotion information? This question applies to most of the observations you listed (e.g., personal experience, cultural or historal significance, nature-centric experience, specific location or city, etc.). This raises my initial question again about what exactly transsuasion is and how much semantic drift appears. Can we really say that transsuasion is "translation" rather than the generation of new content? Furthermore, another question that arises from this is the truthfulness of generated tweets. For instance, if a transsuaded tweet emphasizes a seasonal promotion as a result of transsuasion, this tweet is likely to be hallucinating since no credible information is fed to the model for grounding the transsuasion proccess. Have you measured/considered the truthfulness?

---

> > > ### Author Response · Authors · 2024-11-23
> > >
> > > Thank you for your prompt reply and encouraging response. We are motivated to see that the discussion has been useful in resolving the reviewer's questions and moving the discussion towards a positive outcome. :)
> > >
> > > Please find below the answers to the two questions that you asked:
> > > > While the numbers appear to be high, I am not very clear what is an acceptable number to judge that two tweets have the same meaning and the specified conditions are met. Can you provide some guidance? For example, what are the measurements for the first tweet in Figure 1?
> > >
> > > To provide clarity regarding acceptable similarity metrics for assessing whether two tweets have the same meaning while satisfying the specified conditions, below we give the similarity measurements for all tweet pairs from Figure 1. These metrics include text cosine similarity, text edit similarity, and image cosine similarity (where applicable). For added clarity, we have labeled the similarities and corresponding cases directly in Figure 1 and its caption. Additionally, all transsuasion cases are explained in detail in Appendix Section B.2.
> > >
> > > Further, to help understand semantic similarity thresholds better, we give several qualitative examples below (and in Section D.2), along with their detailed semantic similarity values and explanations. Thank you for indicating this to us. While the paper had detailed quantitative results, we believe this analysis has helped us with improving the qualitative aspects of the paper as well.
> > >
> > > 1. Similarity metrics of the trassuasion tweet pair by the brand “Converse” shown in Figure 1 top-left
> > >   - Case: VisOnly  (Task: Generate an improved image (I2) conditioned on the original image (I1) the original text (T1) and output text (T2). The goal is to generate an improved image while maintaining alignment between the visual content and the textual descriptions)
> > >   - Image Cosine Similarity: 0.77
> > >   - Text Cosine Similarity: 0.71
> > >   - Edit Similarity: 0.25
> > >   - The visual similarity between the original and updated images is moderately high. The new image introduces distinct features (e.g., bold prints and archival designs) while retaining high consistency in product focus (sneakers) and language used.
> > >
> > >
> > > 2. Adobe Example (bottom-left, tweet on Acrobat PDF)
> > >   - Case: Ref  (Task: Change the text so as to increase engagement. The input is content (text) without any media (T1), and the output is improved content (text) without any media (T2). Meaning remains preserved with high semantic similarity in T1 and T2.)
> > >   - Text Cosine Similarity: 0.90
> > >   - Edit Similarity: 0.26
> > >
> > >
> > > 3. GreenPeace example (right, tweet on Bushfires)
> > >   - Case: TextOnly (Transsuade text while the original text (T1) and the original (I1) and output (I2) images are given as input. The output is the transsuaded text (T2). The image (I2) given as input stays constant.)
> > >   - Text Cosine Similarity: 0.92
> > >   - Edit Similarity: 0.26
> > >
> > >
> > > > In Figure 8 in the revision, what model setting does "our performance" refer to? Please clarify in the paper.
> > >
> > > For Figure 8, we select the highest ELO model, i.e., Ours (CS+BS+TS) (13B). Thanks for pointing this out; we have specified this in the figure.

---

> > > > ### Author Response · Authors · 2024-11-23
> > > >
> > > > > I found this analysis very fascinating. For your brand-specific analysis for Bulgari, you mention that transsuaded tweets evoke strong emotional engagement and vivid imagery, emphasize products rather than events, and showcase a unique and innovative design element. Is this conclusion based on a systematic analysis or based on a few examples? The second aspect (i.e., emphasize products rather than events), in particular, seems to somewhat contradict your observation from Starbucks (i.e., emphasize a seasonal theme or promotion). I would expect a more systematic analysis because I believe that can be an important scientific contribution of this work.
> > > >
> > > > > In addition, if a transsuaded tweet becomes more persuasive because it emphasizes a seasonal theme or promotion, does it mean that the original tweet had a different message without this promotion information? e.g., personal experience, cultural or historal significance, nature-centric experience, specific location or city, etc.)
> > > >
> > > >
> > > > Thank you for showing appreciation for the analysis. The analysis was done using the algorithm given below:
> > > >
> > > > **Insights Algorithm**
> > > >
> > > > 1. **Tweet Sampling**: For a brand, we randomly sample 2000 pairs from the Transsuasion (TS) dataset.
> > > > 2. **Insight Generation**: Using GPT-4o-mini, we analyze each tweet pair to generate candidate insights explaining why the more liked tweet (the "transsuaded" tweet) is more engaging. Each tweet pair includes text and image captions, along with their like counts. GPT-4o-mini is tasked to identify potential reasons based solely on the content of the tweets that explain why Tweet 2 performs better than Tweet 1
> > > > 3. **Insight Clustering**: We group similar insights using K-means clustering to consolidate overlapping or redundant insights. This step ensures the resulting set of insights is compact and diverse.
> > > > 4. **Insight Support Calculation and Filtering**: For a given insight (e.g., "Emphasize products rather than events"), we evaluate its support by prompting GPT-4o-mini to classify whether the tweet pair (A,B) align with the insight as follows:
> > > > - **For**: Tweet B follows the insight.
> > > > - **Against**: Tweet A follows the insight.
> > > > - **Neither**: Neither tweet, or both, follow the insight.
> > > > - For example:
> > > > 	- Tweet (A): Originating from the 1960s, this #SerpentiSeduttori heritage watch plays an iconic role in the history of #BulgariWatches. #Bulgari.
> > > >   - Tweet B: The supple bracelet of the #Serpenti Seduttori watch features a hexagonal pattern inspired by the scales of a serpent.  <HYPERLINK> #BulgariWatches #MoreThanAWish #HolidaySeason <HYPERLINK>
> > > > 	- Insight (I): Transsuaded tweets emphasize products rather than events.
> > > > 	- B follows I, while A doesn't, therefore, this pair supports the insight.
> > > >
> > > > 5. **Net Support Score**: We calculate the net support score for each insight as follows:
> > > > - Net Support Score= (For − Against) / Total Evaluations
> > > > - We finally show the top insights here, we will add this to our appendix, the support scores for the Bulgari Insights are given below.
> > > >   - Transsuaded tweets evoke strong emotional engagement and vivid imagery (Net score: 0.19)
> > > >   - Transsuaded tweets emphasize products rather than events. (Net score: 0.13)
> > > >   - Transsuaded tweets showcase a unique and innovative design element. (Net score: 0.11)
> > > >
> > > >
> > > >
> > > > **Starbucks vs Bulgari**
> > > >
> > > > Thank you for your thoughtful observation. Different brands often adopt distinct persuasion strategies tailored to their specific audience and market positioning, which can appear contradictory when compared side by side. For instance, Bulgari, a luxury fashion brand, emphasizes products to align with its aspirational branding. In contrast, Starbucks, operating in the retail food and beverage sector, leverages seasonal themes and promotions to drive customer engagement. This variation underscores the importance of extracting insights at the brand level, as our approach does. Further, by capturing these specific details, our methodology enables the cross-brand learning process, allowing the model to generalize effectively while respecting the unique characteristics of each brand's strategy.
> > > >
> > > >
> > > > **Qualitative Examples of Insights Derived**
> > > >
> > > > The observations (personal experience, cultural or historical significance, nature-centric experience, specific location or city) are the insights derived from the insights algorithm described above. To demonstrate that these insights are grounded in transsuasion pairs that maintain semantic similarity and are not hallucinated, we provide qualitative examples of tweet pairs that support each of these insights below. Each of the examples below consists of the Insight, the original tweet (and image if present), the ground truth transsuaded tweet, the username that posted the tweets, the dates they were posted, and explanations associated with them.

---

> ### Author Response · Authors · 2024-11-23
>
> **Starbucks:**
> 1. Transsuaded tweets emphasize a seasonal theme or promotion.
> - Username: “StarbucksIndia”
> - T1 date: 27th October 2021
> - Tweet: “Relish the sweetness in every sip of the season’s special at your nearest Starbucks store.”
> - T2 Date: 28th October 2021
> - Transsuaded Tweet: “Celebrate #AStarbucksDiwali at your nearest Starbucks store &amp; make this Diwali even more memorable by relishing our season's special with your friends and family at your nearest Starbucks store.”
> - Explanation: “Diwali” is a major festival in India that occurred between 2 November and 6 November in 2021, and happens between October and November every year. The transsuaded tweet uses this fact because of the date in context.
>
>
> 2. Transsuaded tweets convey a personal experience or sentiment.
> - Username: “StarbucksNews”
> - Tweet: “Schultz: Shanghai is the city with the most Starbucks locations in the world <HYPERLINK>”
> - Transsuaded Tweet: “This city has the most @Starbucks locations in the world, your own Shanghai! - Schultz <HYPERLINK>”
> - Explanation: The transsuaded tweet refers to Shanghai as “your own” conveying a more personal experience as opposed to just stating it as a fact.
>
>
> **Airbnb**
> 1. Transsuaded tweets emphasize a nature-centric experience
> - Username: Airbnb Australia
> - Tweet: 📍Tasmania, A 25 minute drive to the stunning Bay of Fires, this two-bedroom home comes complete with a playground and BBQ pit for toasting marshmallows, for the perfect family stay <HYPERLINK> <HYPERLINK>
> - Transsuaded Tweet: 📍Tasmania, This minimalist timber barn sits on three acres at the foothills of the Dial Ranges. Ideal for switching off and slowing down, take your pick of beautiful local climbs and finish the day with a bath under the stars. <HYPERLINK> <HYPERLINK>
> - Explanation: The transsuaded tweet portrays a nature-centric experience with “local climbs, bath under the stars”, while both are used to promote tourism in Tasmania, Australia.
>
> 2. Transsuaded tweets include a specific location or city
> - Username: Airbnb
> - Tweet: Among experience hosts, 50% are women, and 55% of home hosts are women. <HYPERLINK> <HYPERLINK>
> [Image: an Infographic with two women in a kitchen laughing and talking.](https://pbs.twimg.com/media/DXx1O1cU0AATw4y?format=jpg&name=large)
> - Transsuaded Tweet: Women make up a larger majority of home hosts in some countries, like New Zealand (70%), South Africa (63%) and the Philippines (61%). Santa Barbara has the most women experience hosts (65%). <HYPERLINK>
> [Image: an Infographic telling percentage of experience hosts in Santa Barbara are women](https://pbs.twimg.com/media/DXx1O1cU0AATw4y?format=jpg&name=large)
> - Hyperlink: https://news.airbnb.com/women-hosts-have-earned-nearly-20-billion-on-airbnb/
> - Explanation The transsuaded tweet is more engaging due to the use of specific location mentions (both in text and image).
>
>
> **Nike**
> 1. Transsuaded tweets include a specific cultural or historical significance.
> - Username: Nike
> - Tweet: The KOBE XI ‘What The’ combines the highlights into one memorable shoe. <HYPERLINK> <HYPERLINK>
> - Transsuaded Tweet: Legendary shoes for a legendary career. Make your own Kobe XI #MambaDay iD tomorrow. <HYPERLINK> <HYPERLINK>
> - Explanation: The transsuaded tweet leverages the historical significance of Kobe Bryant’s (a prominent Basketball figure) career rather than emphasizing the shoe itself.
>
>
> 2. Transsuaded tweets include a specific date and time.
> - Username: nikebasketball
> - Tweet: Iconic blue for an iconic run. Customize the #KYRIE2. #BringYourGame @NIKEiD <HYPERLINK> <HYPERLINK>
> - Transsuaded Tweet: 20 years that fueled the Love & Hate. #KYRIE2, Customize your own @NIKEiD. <HYPERLINK> <HYPERLINK>
> - Explanation: The transsuaded tweet utilizes specific time period “20 years” which is a common theme of successful tweets for Nike.
>
>
>
>
>
> Thank you again for your valuable feedback, which has improved the qualitative strength and readability of our paper! We hope our answers have satisfied your concerns, and we would love to answer any remaining questions. 🙂

---

> > ### Comment · Reviewer_Xibh · 2024-11-24
> >
> > Thank you for the further analyses. These are all insightful.
> >
> > What is your thought on the truthfulness of transsuaded tweets?
> > This aspect would not completely nullify this work, but I belive this concern is related to ethical issues and should be discussed. What is your observation and can you suggest some methods for mitigating this potential issue?

---

> ### Author Response · Authors · 2024-11-24
>
> Thank you for the follow-up question.
>
> The text of the trassuaded tweet depends on the degree of autonomy given to the LLM. The truthfulness is controlled by the following levers: (semantic similarity, edit similarity, image similarity, link match, NER match, and fact match).
> We have covered these aspects in Table 1 and in the table in our first response. Qualitatively, take the example of this transsuasion pair:
>
> - Tweet: Among experience hosts, 50% are women, and 55% of home hosts are women. <HYPERLINK> <HYPERLINK> Image: an Infographic with two women in a kitchen laughing and talking.
> - Transsuaded Tweet: Women make up a larger majority of home hosts in some countries, like New Zealand (70%), South Africa (63%) and the Philippines (61%). Santa Barbara has the most women experience hosts (65%). <HYPERLINK> Image: an Infographic telling percentage of experience hosts in Santa Barbara are women
> - Hyperlink: https://news.airbnb.com/women-hosts-have-earned-nearly-20-billion-on-airbnb/
>
>
> Here, the human expert picked up a fact different from the one pointed out in the original tweet (from the indicated URL) to make the tweet more persuasive. Similarly, an LLM has to be given some autonomy to change. It can be by picking up a different statement from a set of statements (as in the example above), or just changing the wording and not the core content (Case: Ref), or not changing the tweet text at all, and just changing the image (VisOnly). While an LLM under test for its persuasive capability can change the facts as well to make the content more persuasive, this problem is controlled to a large extent by the way we construct the dataset for transsuasion. As we show in the first reply, GPT-4o (an independent judge) agrees 88.3% of the time that the facts in the transsuasion pair are the same with high confidence. To further enhance the pipeline, future work can look into joining our work, especially PersuasionBench and Arena, with extensive NLP literature on fact-verification.

---

> ### Author Response · Authors · 2024-11-24
>
> After the reviewer's question and our reply yesterday, we conducted more analysis to analyze the truthfulness of the models trained on the transsuasion dataset.
> To understand this better, we computed the performance of trained models with comparable baselines on the TruthfulQA dataset.
>
> TruthfulQA [1] is a standard benchmark included in the widely used LLM benchmark MMLU [2] to measure whether a language model is truthful in generating answers to questions. The benchmark comprises 817 questions that span 38 categories, including health, law, finance and politics. Questions are crafted so that some humans would answer falsely due to a false belief or misconception. To perform well, models must avoid generating false answers learned from imitating human texts [1].
>
>
> This analysis specifically highlights the robustness of trained models with the base LLM and allows us to address potential concerns regarding ethical issues and truthfulness.
> We compare our trained models with their base LLM (lmsys/vicuna-13b-v1.5), Llama-2-chat, and OpenLLaMA [3]—on the MMLU benchmark's TruthfulQA task [1], a well-established benchmark for this purpose. Additionally, we provide scores for the 7B variants of OpenLLaMA, LLaMA-2-chat, and Vicuna to improve the interpretability of the metrics in terms of sensitivity. The corresponding results are presented in the table below, with bar plots provided in Figure 12 of the appendix.
>
>
> | Models| TruthfulQA Accuracy (mc1) | TruthfulQA Accuracy (mc2) |
> |-------|--|---|
> | **lmsys/vicuna-13b-v1.5**| 0.351 ± 0.017             | 0.509 ± 0.015 |
> | **Ours (Instruct)** | 0.348 ± 0.016             | 0.499 ± 0.016|
> | **Ours (CS+BS+TS)** | 0.344 ± 0.016             | 0.495 ± 0.016|
> | **Ours (DPO)**| 0.343 ± 0.016             | 0.492 ± 0.016|
> | **Ours (CS+BS)**                      | 0.335 ± 0.016| 0.488 ± 0.016             |
> | **lmsys/vicuna-7b-v1.5**              | 0.303 ± 0.017| 0.463 ± 0.016             |
> | **meta-llama/Llama-2-13b-chat-hf**    | 0.301 ± 0.016| 0.453 ± 0.016             |
> | **meta-llama/Llama-2-7b-chat-hf**     | 0.280 ± 0.016             | 0.440 ± 0.016             |
> | **openlm-research/open_llama_13b**    | 0.261 ± 0.015             | 0.384 ± 0.014             |
> | **openlm-research/open_llama_7b**     | 0.231 ± 0.015             | 0.351 ± 0.014             |
>
> TruthfulQA measures the performance on the following metrics:
> - MC1 (Single-true): Given a question and 4-5 answer choices, select the only correct answer. The model's selection is the answer choice to which it assigns the highest log-probability of completion following the question, independent of the other answer choices. The score is the simple accuracy across all questions.
> - MC2 (Multi-true): Given a question and multiple true / false reference answers, the score is the normalized total probability assigned to the set of true answers.
>
> All evaluations were done 0-shot, through Harness LM-eval [4]
>
>
> The evaluation confirms that our fine-tuned models maintain truthfulness levels very close to the base model, Vicuna-13b-v1.5, with minimal performance degradation after fine-tuning. Further, our models consistently perform better or comparable to other comparable 13B models, such as LLaMA-2-13B and OpenLLaMA-13B, across both MC1 and MC2 metrics. Importantly, our models preserve the truthfulness of the base LLM while enhancing persuasiveness, validating our approach of using semantically similar training data to maintain factual consistency and truthfulness without compromising on persuasion. We have included this analysis along with the results mentioned in the previous reply in Appendix Section-C.3 of the updated version.
>
>
>
> We hope that we have been able to answer the reviewer's questions and would request the reviewer to increase the score if they agree that the responses have been helpful in addressing the concerns.
>
>
> **References:**
>
> [1] Stephanie Lin, Jacob Hilton, and Owain Evans. 2022. TruthfulQA: Measuring How Models Mimic Human Falsehoods. In Proceedings of the 60th Annual Meeting of the Association for Computational Linguistics (Volume 1: Long Papers), pages 3214–3252, Dublin, Ireland. Association for Computational Linguistics.
>
> [TruthfulQA: Measuring How Models Mimic Human Falsehoods](https://aclanthology.org/2022.acl-long.229) (Lin et al., ACL 2022)
>
> [2] Hendrycks, D., Burns, C., Basart, S., Zou, A., Mazeika, M., Song, D., & Steinhardt, J. (2020). Measuring massive multitask language understanding. arXiv preprint arXiv:2009.03300
>
> [3] Geng, X., & Liu, H. (2023, May). OpenLLaMA: An open reproduction of LLaMA. Retrieved from https://github.com/openlm-research/open_llama
>
> [4] Biderman, S., Schoelkopf, H., Sutawika, L., Gao, L., Tow, J., Abbasi, B., ... & Zou, A. (2024). Lessons from the Trenches on Reproducible Evaluation of Language Models. arXiv preprint arXiv:2405.14782.

---

> > ### Comment · Reviewer_Xibh · 2024-11-25
> >
> > Thank you for the further analysis.
> >
> > While these analyses are useful and it's good to see that the model preserves its truthfulness for general truthful QA, I'm not 100% convinced that transsuaded tweets are factual for the domains covered in this dataset. For instance, the transsuaded tweet provided above contains some statistics, and how can we ensure that these statistics are factual? I hope to see more discussion on this aspect, its potential impact, and some mitigating methods in the final paper.
> >
> > I really appreciate the authors' engagement and additional experiments. I've rasied my score.

---

### Official Review · Reviewer_EsZt · 2024-11-02

**Soundness:** 3
**Presentation:** 1
**Contribution:** 3
**Rating:** 6
**Confidence:** 4

**Summary:**

This paper aims to measure LLM’s persuasiveness. The authors first collect a large-scale dataset from X from corporate accounts where similar messages with difference in details lead to very different levels of engagement. They then conduct experiments focusing on two separate but connected aspects: 1) the ability to simulate: to evaluate the effectiveness of a message 2) to generate: to either generate brand-new messages with the requested account, time and engagement level. They then conduct experiment with a few LLMs, both 0-shot and few shot, as well as fine-tuning a 13B model, which outperform much bigger models. They also conduct experiment showing that even expert humans are not that good in judging how a message would perform.

**Strengths:**

This paper is well-motivated and the authors engage with literature from psychology. The methodology of the dataset collection is very good – the authors make use of similar content posted at different times and the differential engagement level as a signal. Unlike some other recent work in this area, the authors conduct a large amount of validation about some aspects of the paper (e.g. how well do human expert do; human-evaluation of the persuasiveness; how much transfer across domain there is between X post and other domains). The main result (Table 2&3) is clear and convincing. Overall, the scale of this benchmark and the naturalistic setting makes it appealing.

**Weaknesses:**

1) **Conceptual Framework and Claims:**
The paper's framing of "persuasion" is problematic and potentially misleading. The study effectively measures social media post performance prediction and generation, not persuasion as defined in psychological literature. This measurement error also undermines the paper's conclusions about regulation and broader implications. Therefore, I would recommend the authors to reframe the work as a study of social media engagement prediction/generation without overclaiming about persuasion capabilities. Trust me, it is still a valuable paper.

***

2) **Literature Coverage:**
While the level of engagement with some aspect of the literature is good, I am afraid there is some notable omissions, for example, the whole field of computational argumentation from the NLP community, especially argument quality? I would also encourage the authors to engage more with relevant literature when making claims. For example, there are several places in which more citations would be good to have (e.g. Line 417-419, https://arxiv.org/abs/2406.14508; Line 292-295, https://arxiv.org/abs/2404.00750)
***

3) **Structural and Presentation Issues:**
Additionally, the presentation of this paper is somewhat confusing. I had to go through the paper quite a few times and reread things all over the place to grasp what is actually happening.
To be concrete:

a) it would be great if you could have a separate section on related works, rather than embedding it in the intro.

b) it would be great to include a figure of the tasks you are actually measuring (e.g. BS, CS-CT).

c) there is a lack of qualitative result. While the numbers certainty indicates your fine-tuned model drastically lead to higher score in evaluation metrics, could you show paired examples, perhaps in the appendix, as to how exactly the model behave, before vs after fine-tuning, to let the reader contextualize the change in numbers?

d) I am of the opinion that the things starting line 352 (“Extent of Transfer of Persuasive Skills) are more like robustness checks and are not directly relevant to the main benchmark? Perhaps this would then give you enough space to take a deeper dive in the main results, which is somewhat rushed?

e) I think perhaps “observational data” would be a better term to use than “natural experiment”?

f) Please move the ethics section to the main paper (which gives you 1 page that doesn’t take up page limit). The ethics of this study is way too important to be somewhere in the appendix
***

4) **Minor:**
A few minor stylistic things (e.g. line 334, double brackets, try using perhaps square brackets for one pair?)

**Questions:**

**General Question:**

1) Is there any risk of data contamination? I.e. could the LLMs have seen sense tweets as well as their engagement numbers already?

2) Conceptually, where do you think the performance ceiling is? How far can we get in maximizing the persuasiveness of a message?

3) I believe it is still Twitter policy that only Twitter ID can be shared publicly. However, given the exorbitant amount they currently charge for API access to hydrate the tweets, how do you plan on releasing the data you collect?

**Regarding the specific claims in the paper:**

4) Line 296-298, “…measures a model’s capabilities to simulate behavior over (future) time unseen during the training.” But is this future time within LLM’s pretraining knowledge cutoff or not?

5) What is the oracle model in Table 2?

6) Why do you use Vicuna 1.5 13B to finetune?

7) Even though you claim that the time gap between paired posts is not that important (e.g. Table 6), but that table itself seems to show statistically significant correlation for some accounts between like difference and time difference?

8) Line 480-482 “LLaMA-3-70B, while being significantly smaller than GPT-3.5 and GPT-4, has a higher persuasiveness.” – I am afraid I am not following where you get this conclusion?

9) Around line 847-849, “their KPIs. Table 4 shows the results of this study. We
observe that despite being experts in marketing, the budget allocation by these marketers had almost no correlation with any of their key performance indicators.” -> I am not sure if 0.207 is “no correlation”?

10) Line 1431-1437: I think T1 and T2 here are actually semantically quite different? Could you mention how many pairs are like this where one version of the post actually contains much more information than the other one? This undermines your data collect regarding “matching” posts, right?

11) Line 1545-1555: Do you think it would be better if you define what low/medium/high actually means for the LLM in the prompt? I think perhaps there’s a potential of underestimating 0-shot model performance due to this?

---

> ### Author Response · Authors · 2024-11-22
> **Thank you for your thoughtful review**
>
> > Conceptual Framework, The paper's framing of persuasion
>
> We appreciate the reviewer's concern and would like to address the distinction between persuasion and social media engagement in the context of our work.While it's true that our primary dataset leverages social media interactions, we position our work within the domain of persuasion due to its broader transferability and application beyond engagement metrics.
>
> To check the transfer of persuasion capabilities measured over Twitter to other domains, we test all models on 4 benchmarks in zero and few shot: Humans as Judge (our study), Humans as Judge (Anthropic Persuasion study), Marketing blogs dwell time and views prediction, Audience specific transcreation. This also allows us to test the transfer power of Twitter-finetuned 13B model whose persuasion capabilities were developed over Twitter to other channels and domains.
> Marketing blogs dwell time and views prediction: We improve 19% and 22% compared to base model on dwell time and views prediction respectively (refer to table 9)
> Audience specific transcreation: We improve the performance ~2x compared to the base model on targeting (refer to Table 10)
> Humans as judges (our study): we improve by 15%, 20% and 50% on on Upvote/Downvote classification, Reasoning classification, and Feedback perplexity compared to the base model (refer to table 7)
> Humans as judge (Anthropic study): Our rank correlation is 0.47, ~6.5x more compared to the base model (0.07) (refer to table 8)
> Further to measure the robustness of PersuasionArena we collect the rankings from all metrics (8) of the (4) transfer benchmarks above, next we calculate the mean kendall tau coefficient between PersuasionArena these 8 rankings, this gives us a mean Kendall Tau coefficient of 0.69 which is considered strong. The consistent rankings of PersuasionArena in zero/few shot and unseen domains further validate its importance.
> These show that models trained on the task of transsuasion also transfer to completely unseen domains, channels, behaviors, and tasks. These findings demonstrate the robustness of our framework and the broader applicability of PersuasionArena and Bench across varied tasks involving persuasion. The term "persuasion" captures the transfer capabilities demonstrated by our framework, distinguishing it from just social media engagement prediction. The consistent generalization of PersuasionArena across varied studies [1,2,3] which frame this problem statement as “Persuasion” validates the broader implications of our work.
>
> > Literature Coverage
>
> Literature Coverage: Prior research in argument quality highlights features like reason-giving, logical coherence, originality, relevance, interactivity, logos, ethos, and pathos as key to persuasive arguments [4,5,6]. For instance, arguments that clearly explain reasons or use credible evidence, such as statistics or sources, are often rated as more convincing. Logical coherence, achieved through clear and relevant progression, also plays a vital role. Annotators tend to prefer succinct and focused arguments, especially when comparing pairs, as these maintain clarity and avoid off-topic points.
>
> Our work on transsuasion aims to improve measure and improve persuasion in language, while keeping the semantic meaning intact. We will revise our related work section to clearly position transsuasion within this and explain how it complements and extends these foundational contributions. Thank you for helping us better articulate our work.
>
> In their research, Hackenburg et al. (2024) identify a log scaling trend for political persuasion with large language models, showing that persuasiveness gains diminish as model size increases. This study provides evidence that beyond task completion capabilities, scaling model size may have limited impact on enhancing persuasive effectiveness. Thank you for the valuable literature, we have added them in lines 195 and 457 of the current version.

---

> ### Author Response · Authors · 2024-11-22
>
> **Structural and Presentation Issues**
>
> > it would be great if you could have a separate section on related works, rather than embedding it in the intro.
>
> Thanks for your suggestion we have added a related work section in the appendix (section G) we will separate and add these in the camera ready version of the paper.
>
> > it would be great to include a figure of the tasks you are actually measuring (e.g. BS, CS-CT).
>
> We have included the task figure(Figure 11 in the new version) in the Appendix.
>
> > there is a lack of qualitative result. While the numbers certainty indicates your fine-tuned model drastically lead to higher score in evaluation metrics, could you show paired examples, perhaps in the appendix, as to how exactly the model behave, before vs after fine-tuning, to let the reader contextualize the change in numbers?
>
> We agree with your suggestion, we have improved the presentation and added examples for transsuasion from the untrained model in Section D. Notable trends like “usage of urgency, complex design elements” are missed which we have validated through our analysis in response to reviewer Xibh,  What makes content persuasive
>
> > I am of the opinion that the things starting line 352 (“Extent of Transfer of Persuasive Skills) are more like robustness checks and are not directly relevant to the main benchmark? Perhaps this would then give you enough space to take a deeper dive in the main results, which is somewhat rushed?
>
> The goal of our work is to build a comprehensive framework to measure the persuasiveness of LLMs, motivated by benchmarks measuring other natural language capabilities: BigBench, HELM, MMLU, MSR-VTT, CodeXGLUE, and MATH. We believe that it is very important to measure persuasiveness on a breadth of tasks covering different aspects including predictive and generative capabilities, which is why we created a meticulous benchmark to cover the generalization across 6 (BS, CS, TS-CT, TS-GT, TC, and Hum-Per) tasks.
>
> > I think perhaps “observational data” would be a better term to use than “natural experiment”?
>
> Natural Experiments and Observational Studies are used interchangeably in literature [6,7], we understand the confusion and will clarify this in the camera ready version of the paper.
>
> > Please move the ethics section to the main paper (which gives you 1 page that doesn’t take up page limit). The ethics of this study is way too important to be somewhere in the appendix
>
> Thank you for the valuable suggestion. We agree with this, and will shift the Ethics section to the main paper in the camera ready version.
>
> **Questions**
>
> > Is there any risk of data contamination? I.e. could the LLMs have seen sense tweets as well as their engagement numbers already?
>
> The simulation results for all zero/few shot elements in Persuasion Table 2, with nearly random performance largely indicates that this should not be the case.
> Further our time split consists of samples from the latest months, which is outside the knowledge cutoff window for LLaMA-2 and we found no such examples in Vicuna SFT datasets either.
>
> > Conceptually, where do you think the performance ceiling is? How far can we get in maximizing the persuasiveness of a message?
>
> Thank you for posing such an intriguing question.
> Based on our benchmarks, the persuasiveness of large language models (LLMs) generally falls short of the best human writers. However, in specific domains and topics, their performance can sometimes surpass even top human writers. This advantage stems from LLMs' ability to aggregate and learn from a vast and diverse pool of data, far exceeding the exposure any single human writer could achieve.
> As the scale of available data grows, we anticipate that these models' performance will continue to improve. However, it's important to note that societal norms and perceptions of persuasiveness evolve over time. For example, messaging that resonated strongly during the COVID-19 era might not have the same impact today. Similarly, trends often reemerge—consider how retro aesthetics or communication styles periodically make a comeback.
> This fluidity means that no single model can remain the ultimate persuader indefinitely. Instead, it’s likely to resemble a "police-and-robber" dynamic, with models striving to keep pace with societal shifts while simultaneously influencing those shifts by introducing new trends.
> Ultimately, this underscores the importance of ongoing benchmarking and continuous training to ensure that models remain effective and relevant over time.

---

> ### Author Response · Authors · 2024-11-22
>
> > I believe it is still Twitter policy that only Twitter ID can be shared publicly. However, given the exorbitant amount they currently charge for API access to hydrate the tweets, how do you plan on releasing the data you collect?
>
> We plan to open-source the collected datasets, source code, and trained models.
> Our aim is to build PersuasionArena and PersuasionBench, inspired by LMSys. LMSys is supported through public donations (https://lmsys.org/donations/) from organizations such as Hugging Face, MBZUAI, Kaggle, NVIDIA, and others. Similarly, a few organizations have already reached out privately, offering resources to support the development of open-source infrastructure for PersuasionBenchand PersuasionArena. To benchmark their models on LMSys, model developers submit either model parameters or APIs. Following a similar approach, we plan to run benchmarks and arenas where model developers can continuously evaluate their models. These resources will be released upon acceptance.
> Regarding the release of actual Twitter data, you are correct—Twitter IDs can be publicly shared. We will therefore release the tweet IDs so developers with Twitter API access can begin working on the benchmarks. For those without access, the last time we reviewed Twitter's Developer Terms and Conditions (in January 2023), it was permissible to share datasets with fewer than 1 million tweets. Since then, the terms have been updated. This 1-million constraint still covers the majority of our samples (>90% of transsuasion data). We plan to confirm with the Twitter API team whether this constraint remains valid.
>
> > Line 296-298, “…measures a model’s capabilities to simulate behavior over (future) time unseen during the training.” But is this future time within LLM’s pretraining knowledge cutoff or not?
>
> LLaMA-2’s cutoff date is earlier than our time split, and we found no tweets in the SFT dataset of Vicuna-1.5
> What is the oracle model in Table 2?
> The Oracle is a Vicuna-1.5-13B (same as the base LLM for Transsuasion) finetuned on Train+Test data, the details are outlined in lines [342-353] of the current version, We will fix the typo in line [346] of the current version. The oracle is trained on the train and test data with the best training regime (for the comparative transsuasion tasks) to measure the persuasiveness of the transsuaded tweet.
>
> > Why do you use Vicuna 1.5 13B to finetune?
>
> At the time we were conducting the experiments, Vicuna 1.5 13B was the most capable model (in the 13B parameters scale), and had a knowledge cutoff before our time-split.
>
> > Even though you claim that the time gap between paired posts is not that important (e.g. Table 6), but that table itself seems to show statistically significant correlation for some accounts between like difference and time difference?
>
> Yes, we also agree that there is a statistically significant relationship, but it is very small (-0.079 to 0.087), we have rephrased it to “very small” to make it clearer.
>
> > Line 480-482 “LLaMA-3-70B, while being significantly smaller than GPT-3.5 and GPT-4, has a higher persuasiveness.” – I am afraid I am not following where you get this conclusion?
>
> Thank you for pointing out, GPT-4 is a typo there, we have fixed it in the current version at line [518-519]
>
> > Around line 847-849, “their KPIs. Table 4 shows the results of this study. We observe that despite being experts in marketing, the budget allocation by these marketers had almost no correlation with any of their key performance indicators.” -> I am not sure if 0.207 is “no correlation”?
>
> Thank you for pointing out, we have corrected it to “low or no correlation” in table 4.
>
> > Line 1431-1437: I think T1 and T2 here are actually semantically quite different? Could you mention how many pairs are like this where one version of the post actually contains much more information than the other one? This undermines your data collect regarding “matching” posts, right?
>
> The example you have mentioned belongs to the Highlight case (the hyperlink mentioned is same and LLM’s are given the context of the webpage, for details please refer to  Hilight in Section B.2 of the appendix) both tweets utilize different information from the webpage, which require a lower level of cosine semantic similarity in text (>0.6)
>
> > Line 1545-1555: Do you think it would be better if you define what low/medium/high actually means for the LLM in the prompt? I think perhaps there’s a potential of underestimating 0-shot model performance due to this?
>
> Thank you for pointing this out, the listing that you have mentioned was used to train the model, we have added the listings for 0-shot evaluation in Listing 25, we tried several strategies including rating out of 3, top, bottom and middle, but this worked the best in our empirical evaluation.

---

> > ### Author Response · Authors · 2024-11-22
> >
> > [1] Durmus, E., Lovitt, L., Tamkin, A., Ritchie, S., Clark, J., & Ganguli, D. (2024, April 9). Measuring the Persuasiveness of Language Models. https://www.anthropic.com/news/measuring-model-persuasiveness.
> >
> > [2] Gpt-o1 system card." https://openai.com/index/openai-o1-system-card/
> >
> > [3] Hancock et al, AI-Mediated Communication, JCMC, 2019
> >
> > [4] Zhongyu Wei, Yang Liu, and Yi Li. "Is This Post Persuasive? Ranking Argumentative Comments in Online Forum." Proceedings of the ACL, 2016​.
> >
> > [5] Ivan Habernal and Iryna Gurevych. "Which Argument is More Convincing? Analyzing and Predicting Convincingness of Web Arguments Using Bidirectional LSTM." Proceedings of the ACL, 2016
> >
> > [6] Dunning, Thad. Natural experiments in the social sciences: A design-based approach. Cambridge University Press, 2012.
> >
> > [7] Natural Experiment (Wikipedia) https://en.wikipedia.org/wiki/Natural_experiment

---

> > > ### Comment · Reviewer_EsZt · 2024-11-24
> > >
> > > Thank you very much for your responses and I really appreciate the thoroughness. I am raising the score by 1.
> > >
> > > Some remaining comments:
> > >
> > > On conceptual framework and framing:
> > >
> > > Unfortunately, I am not convinced - I am asking a conceptual question that has little to do with experimental results. Just because you tune on a task and that seems to improve model performance on a persuasion study doesn't make your task persuasion. Imagine if I fine-tune on the world value survey and then the model performance on these 4 benchmarks improve, does that make WVS a persuasion benchmark?
> > >
> > > I would like to remind you that multiple reviewers pointed out the difficulty of navigating this paper and I would appreciate a major overhaul of the writing whether this paper is accepted here or in another venue.

---

> > > > ### Author Response · Authors · 2024-11-30
> > > >
> > > > > On Conceptual Framework and Framing
> > > >
> > > > Thank you for your suggestion. While we initially framed this work as focusing on persuasion due to its generalizability across different domains, we have reconsidered and will update the framing to emphasize engagement in the camera-ready version of the paper.
> > > >
> > > > Regarding your conceptual question, our goal was to demonstrate PersuasionArena’s broad applicability on unseen persuasion benchmarks through a diverse range of tasks. This was not solely about fine-tuning but about evaluating transfer across domains. Extensive experiments (Tables 7-12) evaluated PersuasionArena across varying human preference types, communication channels, KPIs, and content types. The results show a strong average rank correlation of 0.69 with unseen persuasion tasks, highlighting that PersuasionArena serves as a robust indicator of LLM persuasiveness across a wide spectrum of content and industry sectors.
> > > >
> > > > For example, as shown in Table 3, GPT-4o outperforms GPT-4 by 38 Elo points and GPT-3.5 by 160 Elo points on PersuasionArena, corresponding to win rates of 55.45% and 71%, respectively, based on Bayes-Elo calculations. These results align closely with the OpenAI GPT-o1 model card [1], which reports a 78.1% win rate on the ChangeMyView benchmark (compared to 71% on PersuasionArena). Notably, this independent validation by OpenAI underscores the reliability of PersuasionArena’s oracle rankings (GPT-o1 scorecard had not been released at the time of our study).
> > > >
> > > > We again appreciate your feedback and concur that refining the conceptual framing of the work from persuasion to engagement will strengthen its clarity.
> > > >
> > > >
> > > > > ​​the difficulty of navigating this paper
> > > >
> > > > We are completely committed to improving the writing of the paper, we are listing below the changes we have already made and we will refine or add in the camera ready version of the paper to improve the organization and presentation.
> > > > - **Dataset Topic Analysis** We have added the visualization for the sectors, industry of our brands, and domains and topics of PersuasionBench in Figure 5 and 6 respectively. We have also added analysis of linguistic features such as emojis, sentiment, and hashtags (in figure 7)
> > > > - **Brand Specific insights from Transsuaded Tweets** We have added some insights into what is more persuasive for a few brands by clustering pairs of tweets from brands and summarization of trends. We calculate the support for our insights and show the top insights in appendix section B.5.
> > > > - **Qualitative Examples of Insights**:  We provide qualitative examples of tweet pairs that support each of the insights. Each of the examples below consists of the Insight, the original tweet (and image if present), the ground truth transsuaded tweet, the username that posted the tweets, the dates they were posted, and explanations associated with them.
> > > > - **Win rate across topics**: In Figure 8, we present our highest and lowest average win rates against GPT-3.5, GPT-4, and GPT-4o across various topics. Our performance reaches its peak, with win rates close to 80%, on topics such as retail product innovation, e-commerce sales, and new launches. Conversely, on topics where we have the lowest win rates such as market analysis, platform growth, and fashion in tech GPT-4o performs comparably or slightly better.
> > > > - **Improved listings for qualitative samples**: We have added dataset, generation, and topline samples and have improved their readability and visualization.
> > > > - **Related work**: We have added a related work section in the appendix (section G). In the camera ready version of the paper we will separate the introduction and related work, to improve the current structure. We have also expanded this section to include more literature on very recent works in persuasion, argument quality, and negotiation with LLMs.
> > > > - **Task Organization**: To improve the readability and organization of these tasks and subsequent results and discussions, we have added a figure (figure 11) illustrating different transsuasion tasks for improved clarity in the appendix
> > > > - **Ethics Section**: We recognize the importance of the ethical considerations in our work, and keeping this in mind we will move the Ethical implication section H from the appendix to the main paper where it is highlighted better for the readers.
> > > > - **Analysis of Dataset Curation Steps**: We have added a detailed step by step analysis of the impact of each individual filtering step on the dataset quality.
> > > >
> > > >
> > > >
> > > > **If we have been able to address the reviewer's concerns and suggestions, we would sincerely appreciate if they could increase the score to acceptance. Thank you for engaging with us to improve the paper!**

---

### Official Review · Reviewer_UfKL · 2024-11-04

**Soundness:** 2
**Presentation:** 1
**Contribution:** 2
**Rating:** 3
**Confidence:** 3

**Summary:**

The authors introduce a dataset and benchmark for measuring Persuasion. Both PersuasionArena and PersuasionBench measure _transsuasion_, a task that involves independently increasing a message’s persuasiveness without changing the underlying content or message. The authors source their dataset through a set of naturally occurring tweets that differ in likes. Finally, the authors outline results across a large range of both generative and predictive tasks; and train models to identify and generate transsuasive text.

**Strengths:**

The paper has several strengths. The authors

1. carefully evaluate a range of models—across both vision and language.
2. leverage online communities and real-world feedback to curate a dataset.
3. will release a dataset of minimally different tweets, alongside metadata and an evaluation setup.

**Weaknesses:**

My first concern is that small changes in the input can yield significantly different meanings that cosine similarity or edit distance may not capture.

Consider the Converse example in Figure 1: using an entirely different _brand_ is probably more likely to have a big effect on persuasiveness—but not because of something the writer can control. I worry that the “minor tweet edits” often have larger semantic variation: a single brand name will significantly affect popularity (as a more extreme example, a model optimizing persuasivness might always prefer to generate “iPhone” or “Andriod”—even if the brand or seller can only sell one or the other)

My second concern has to do with some clarity on timing. It may be that certain times are just more active. Twitter has a cyclic activity pattern that peaks at specific times. This likely has a confounding effect on tweets that appear even hours apart. How did the authors control for these confounds? I see that the authors did some analysis in Table 5; and that some of the p-values are indeed < 0.05. How exactly was this analysis done? I'm trying to interpret the correlation coefficient.

My final concern has to do with general clarity: I found it a bit challenging to navigate the tasks outlined in the paper. I generally feel like there was more emphasis on creating many tasks instead of focusing on a few very important ones. More generally, I worry that focusing on many tasks detracts from the contributions of the dataset. For example: clustering the tweets, running some kind of topic analysis, or highlighting the types of domains the dataset consists (in the main paper) would provide more value than an additional task.

I also have a handful of concerns formulated as questions below:

**Questions:**

A handful of questions can be addressed alongside the weaknesses.

Why call (1) simulative and not predictive? I feel like prediction and generation are more natural contrasts. In general, it was a bit difficult to parse through the tasks in S3. I think a segmentation between tasks that require classification v.s. generation would help clear up the structure. For example, I see why 2.2 is under generative capabilities—but seeing “simulation” in the name may cause unnecessary confusion.

Relatedly, do the authors mean NLG evaluation (line 322) instead of NLP evaluation? Where G = generation?

Minor: I think NegotiationBench has a lot to do with persuasion.
https://arxiv.org/pdf/2402.05863 may be relevant.

Dataset and Error Analysis: While the authors spend some time outlining tasks and evaluating models, I would’ve rather preferred a topic analysis on the kinds of tweets in the dataset—to get a feel for what is “more persuasive” across specific domains.

How should I interpret the topline not being the “best” model? Are the models generating samples with “super-human” persuasion? Or is there some test leakage given the oracle was trained on the full dataset? I think an analysis of what the samples look like in this case would be great.

A final question (minor question): the setup of T1, T2 pairs reminds of an RLHF-esque setup. Why didn’t the authors pursue a “reward model” paradigm?

---

> ### Author Response · Authors · 2024-11-21
> **Thank you for your review.**
>
> We sincerely thank the reviewer for their feedback. Below, we try to address the reviewer's questions and concerns.
>
> > Using an entirely different brand is probably more likely to have a big effect on persuasiveness
>
> We only consider tweet pairs from the same “brand”, more specifically, the sub-brand (e.g. AdobePhotoshop, AdobeLightroom, GoogleChrome, GoogleAI) or twitter username (mentioned in Line [235]). Other than having the same brand, the samples should be less than 45 days apart and are semantically and lexically close to each other.
>
>
> > Small changes in the input can yield significantly different meanings that cosine similarity or edit distance may not capture
>
> > “minor tweet edits” often have larger semantic variation
>
> > “iPhone” or “Andriod”
>
> While it is true that tweets from individual users can vary in almost all aspects of the message (e.g. Android and Iphone), considering that problem itself, we keep our focus on enterprise accounts. The products and services of such accounts remain relatively consistent over time. This consistency allows marketers to experiment with various messaging strategies, resulting in differential audience engagement rates. Further, since the products stay the same, it limits the autonomy of the marketer and consequently the transsuasion data and the models trained on it.
> The question about transsuaded text having similar meaning pertains to the degree to which AI can operate on messages without the sender’s supervision. This could be considered as “autonomy” of AI agent ([3] Hancock et al). We define nine types of transsuasion (Table-1) varying the autonomy of the AI agent. E.g. autonomy can be restricted to changing just the image (VisOnly), or just paraphrasing text (Parap).
>
> To verify this quantitatively, we evaluated the generations on the task “NER Match”, which covers Organizations and Products. We see that our model correctly uses 92% of the NERs, similar to the few shot base model. Results for this are given below.
>
>
>
> In the table below (also added as Table 15 in the Appendix), we show the similarity metrics between transsuaded tweets obtained or generated from (1) ground truth GT (2)  GPT-4o (3) Ours(13B) (4) Vicuna(13B). We use the following metrics to test semantic similarity between transsuasion pairs of the ground truth and generated samples:
>
> 1. NER Match: The number of named entities that match on both sides divided by total named entities
> 2. Factuality match: We prompt GPT-4o to assess if both tweets do not differ factually, we also ask it to give a confidence rating on a scale of 5 tell if it is sure about both the facts being different and only consider the pairs where the confidence is at least 4.
> 3. MetricsMatch: The proportion of pairs of tweets that follow all the constraints for that task (edit similarity, text similarity, and image similarity) as defined in Table 1
>  - PARAP: Text Edit Similarity > 0.6, Text Cosine Similarity > 0.6
>  - REF: Text Cosine Similarity > 0.8
>  - VisOnly: Image Cosine similarity > 0.7
>
> | Model\Metrics                 | **NER Match** | **Factuality Match** | **MetricsMatch** |
> |-------------------------------|---------------|-----------------------|-------------------|
> | **GPT-4o**                    | 97.8%         | 94.1%                | 87.6%            |
> | **Vicuna (13B)**              | 92.7%         | 84.2%                | 80.1%            |
> | **Ours (13B)**                | 92.1%         | 93.6%                | 85.2%            |
> | **GT**                        | 87.1%         | 88.3%                | -                |
>
> The tables show that not only are the input pairs largely similar semantically, but also that the output pairs are more controlled.
>
>
> > I see that the authors did some time analysis in Table 5; and that some of the p-values are indeed < 0.05. How exactly was this analysis done? I'm trying to interpret the correlation coefficient.
>
> Table 5: In Table 5, we calculate and show the correlation between (1) the difference between the time of posting of two tweets and (2) the difference between the likes of these tweets. A negative correlation would imply that earlier or older tweets get more likes. We find that all significant correlations (across all tasks) are negligible (-0.05 to  -0.006). This implies that there is very little impact of the time difference in the difference of likes between the pairs of tweets (across all tasks).

---

> ### Author Response · Authors · 2024-11-21
>
> > clarity on timing. It may be that certain times are just more active. Twitter has a cyclic activity pattern that peaks at specific times. This likely has a confounding effect on tweets that appear even hours apart. How did the authors control for these confounds?
>
>
> Cyclic engagement To analyze the relationship between tweet success (\(Y\)) and temporal features (\(X_1\): hour of the day, \(X_2\): day of the week), while accounting for the brand (\(Z\)). We began with paired tweets (\( Tweet_X, Timestamp_X \)) and (\( Tweet_Y, Timestamp_Y \)), where \( Tweet_Y \) is always more persuasive (e.g., higher likes) than \( Tweet_X \). Each pair was converted into independent samples: \(Tweet_X \) was labeled \(Y = 0\), and \( Tweet_Y \) was labeled \(Y = 1\), retaining their corresponding temporal features and brand identifiers.
> We stratified the data by \(Z\) (brand) to account for brand-specific differences in tweet performance, ensuring that temporal effects were evaluated independently of brand.
>
> For each brand, we performed a chi-square test of independence between \(X_1/X_2\) and \(Y\), calculating individual p-values and effect sizes (Cramér’s V). Brands with fewer than 500 samples were excluded to avoid unreliable results (the coverage still remains > 95%). Finally, we combined p-values using Fisher’s method and calculated a weighted average of Cramér's V across brands to quantify the overall effect.
> The results showed a statistically significant relationship between temporal features (\(X_1/X_2\)) and tweet success (\(Y\)), with a combined p-value of \(p < 0.05\). However, the weighted average of Cramér's V indicated that the effect size was small to very small (e.g., \(0.05 <= Cramér's V < 0.1\)). This suggests that while the time of day and day of the week have a measurable effect on tweet performance, the magnitude of the effect is limited.
>
> Specifically, the weighted average of Cramér's V was **\(0.06\) for \(X_1\) (hour of the day) and \(0.09\) for \(X_2\) (day of the week), suggesting a limited magnitude of influence**. Despite the small effect sizes, the relationship is statistically significant, with a combined p-value of \(1.345 * 10^{-5}\). This indicates that the time of day and day of the week have a measurable but negligible effect on tweet performance as shown in other studies [1,2].
>
> Our original hypothesis was that these effects would be reduced because we only consider brand accounts, which to some extent optimize their timings already. We find that the p-values from the chi-square tests are negatively correlated (-0.31, p=0.003) with the number of followers a brand has, this indicates that more successful brands observe lesser effect of timing (hours of day and days of week) on their engagement. Thanks for the suggestion, we will add this analysis to the paper.
>
> 1. Rea, Louis M., and Richard A. Parker. Designing and conducting survey research: A comprehensive guide. John Wiley & Sons, 2014.
> 2. Gignac, Gilles E., and Eva T. Szodorai. "Effect size guidelines for individual differences researchers." Personality and individual differences 102 (2016): 74-78.
> 3. Hancock et al, AI-Mediated Communication, JCMC, 2019
>
>
> > My final concern has to do with general clarity: I found it a bit challenging to navigate the tasks outlined in the paper. I generally feel like there was more emphasis on creating many tasks instead of focusing on a few very important ones. More generally, I worry that focusing on many tasks detracts from the contributions of the dataset. For example: clustering the tweets, running some kind of topic analysis, or highlighting the types of domains the dataset consists (in the main paper) would provide more value than an additional task.
>
> The goal of our work is to build a comprehensive framework to measure the persuasiveness of LLMs, motivated by benchmarks measuring other natural language capabilities: BigBench, HELM, MMLU, MSR-VTT, CodeXGLUE, and MATH. We believe that it is very important to measure persuasiveness on a breadth of tasks covering different aspects, including predictive and generative capabilities, which is why we created a meticulous benchmark to cover the generalization across 6 (BS, CS, TS-CT, TS-GT, TC, and Hum-Per) tasks.
>
> However, we agree with the suggestion of adding dataset analysis; therefore, we have included additional details, such as dataset statistics, industry distribution of brands, and topic analysis of tweets from each industry. To understand the industry distribution of brands in our dataset, we extracted topics from their Twitter bio using BERTopic. These topics were clustered and assigned a name by GPT-4o-mini  and visualized using a sunburst graph (**Figure 5** in the new version of the paper). Similarly, we also analyzed the topic distribution of tweets (**Figure 6** in the new version of the paper).
> If there is any additional analysis that will highlight our contributions better, please let us know.

---

> > ### Author Response · Authors · 2024-11-21
> >
> > > Why simulate and not predict?
> >
> > We named it following recent literature in this domain that refers to this task as simulation. [4,5,6]
> >
> > > Segmentation between tasks that require classification v.s. Generation
> >
> > Regarding line 322, thanks for the correction, we will change it to NLG evaluation.
> >
> >
> > > Minor: I think NegotiationBench has a lot to do with persuasion. https://arxiv.org/pdf/2402.05863 may be relevant.
> >
> > Negotiation Bench: Our work significantly differs from negotiation-bench in the following aspects. However, it is an important citation for our related work, and we will add it to the paper.
> >
> > - It involves three tasks (Price negotiation, shared resources, and aggregate resources), which focus on utilitarian tasks (synthetic environment), as opposed to more real-world scenarios like social media, speech, etc.
> > - No real human data was used in their analysis. Rather. NegotiationBench tries to simulate human behavior by GPT-4 prompting.
> > Their limitations further illustrate this: for e.g., NegotiationBench shows that GPT4 does not act as an irrational agent. On the other hand, it is widely known that humans act as irrational agents (Kahneman ‘79).
> > - Aristotle, in his seminal work on rhetoric, divides persuasion strategies into pathos (emotion), ethos (credibility and authority), and logos (logic). NegotiationBench uses only logic (logos) for negotiation while ignoring pathos and ethos. We do not create such divides and use all the natural strategies used in real human interactions.
> > - Over time, persuasion strategies evolve. For example, strategies that might have worked during the COVID-19 pandemic a few years ago may not be applicable now. Our mechanism is extensible and automated to adapt to changing trends.
> > - Our measurement shows the transfer of persuasion across tasks and domains like Transcreation, Humans-as-judge of persuasiveness and Simulating the KPIs for a Fortune-500 company’s marketing blogs. On the other hand, NegotiationBench does not show any transfer of negotiation capabilities.
> >
> >
> > > Dataset and Error Analysis: While the authors spend some time outlining tasks and evaluating models, I would’ve rather preferred a topic analysis on the kinds of tweets in the dataset—to get a feel for what is “more persuasive” across specific domains.
> >
> > Thank you for the suggestion. Below, we have added some insights into what is more persuasive for a few brands.
> > To extract these brand-specific insights, we first cluster the embedded (using RoBERTa) transsuasion pairs of a brand, then for each cluster, we ask GPT-4o-mini to summarize the insights from a cluster. We have added them below (also in appendix section B.5)
> >
> > - Bulgari
> >   - Transsuaded tweets evoke strong emotional engagement and vivid imagery
> >     - Example: https://x.com/Bulgariofficial/status/1856736301657235947
> >   - Transsuaded tweets emphasize products rather than events.
> >     - Example: https://x.com/Bulgariofficial/status/1843573678736584907
> >   - Transsuaded tweets showcase a unique and innovative design element.
> >     - Example: https://x.com/Bulgariofficial/status/1846936730471129102
> >
> > Similarly, we have extracted rules for some more brands and industries below
> > - Starbucks
> >   - Transsuaded tweets emphasize a seasonal theme or promotion.
> >   - Transsuaded tweets convey a personal experience or sentiment.
> > - Nike
> >   - Transsuaded tweets indicate urgency or time sensitivity
> >   - Transsuaded tweets include a specific date and time.
> >   - Transsuaded tweets emphasize a specific cultural or historical significance.
> > - Airbnb
> >   - Transsuaded tweets evoke a nature-centric experience.
> >   - Transsuaded tweets emphasize a specific location or city.
> >   - Transsuaded tweets emphasize local economic impact.
> >
> > > How should I interpret the topline not being the “best” model?
> >
> > The topline is the collection of the top 75% of tweets for every brand in their bimonthly period. We find that a trained model is able to generate tweets better than the top 75th percentile. The trained model is learning what persuasion strategies work across different brands across different industry. The data scale across industries and brands is causing the trained model to have a better persuasion capability than the human topline of top 75 percentile.
> >
> >
> > To analyze this trend further, we extracted the brands where the trained model has the maximum improvement over the human topline. We found that the correlation of samples that improve over the top line of a brand with the number of followers of the brand is -0.38 (p=0.0067). The correlation shows that a part of the observed super-human persuasion can be explained by the better performance of the trained model on smaller brands.

---

> ### Author Response · Authors · 2024-11-21
>
> > A final question (minor question): the setup of T1, T2 pairs reminds of an RLHF-esque setup. Why didn’t the authors pursue a “reward model” paradigm?
>
> Reward model paradigm: Thanks for the suggestion, we have added details for our ablation using DPO in PersuasionArena  (extended in Table 3). The DPO model gets an ELO of 1283 compared to the ELO of SFT, which is 1304. While DPO shows lesser ELO than the SFTed model, it shows higher performance on similarity metrics: semantic similarity (87.2%), factuality match (94.3%), and NER match (94.9%). Our primary motivation for doing SFT as opposed to DPO is that models from prior works [7] trained to generate performant content have benefitted from multi-task training (Content Simulation, Comparative Transsuasion) (Please refer to appendix, table 8 in [7])
>
>
> [4] Xie, Chengxing, et al. "Can Large Language Model Agents Simulate Human Trust Behaviors?." (Neurips 2024)
>
> [5] Aher, Gati V., Rosa I. Arriaga, and Adam Tauman Kalai. "Using large language models to simulate multiple humans and replicate human subject studies." International Conference on Machine Learning. PMLR, 2023.
>
> [6] Chen, Weize, et al. "Agentverse: Facilitating multi-agent collaboration and exploring emergent behaviors." The Twelfth International Conference on Learning Representations. 2023.
>
>  [7] Harini, S. I., et al. "Long-Term Ad Memorability: Understanding & Generating Memorable Ads." (WACV)

---

> > ### Author Response · Authors · 2024-11-22
> >
> > Dear Reviewer,
> >
> > We hope our response has fully addressed your concerns. Please let us know if there is anything further we can do to ensure the paper meets the acceptance threshold.
> >
> > Cheers

---

> ### Author Response · Authors · 2024-11-24
>
> Dear Reviewer UfKL,
>
> We trust our response has addressed your concerns and provided the necessary clarity. We kindly wish to note that other reviewers, such as EsZt, have increased their scores after their concerns were addressed, and reviewer Xibh has promised to increase their score following their last comment. We sincerely believe their helpful engagement has increased the paper's quality. If there’s anything further we can do to enhance the paper and ensure it meets the acceptance criteria, please don’t hesitate to let us know.
>
> Thank you!

---

> ### Comment · Reviewer_UfKL · 2024-11-25
> **Thanks for the response**
>
> Thanks for the comprehensive response! I am leaning towards keeping my score-
>
> **Regarding content confounds:**
>
> First, even within the same _brand_ or sub-category may often have tweets that focus on a specific feature. So for photoshop, my tweet might end up focus on two different features within Photoshop, confounding persuasiveness. Do the authors have an anonymous dataset link I can look through? I'd love to take a look at some samples myself.
>
> **Regarding topline generations:**
>
> > To analyze this trend further, we extracted the brands where the trained model has the maximum improvement over the human topline.
>
> Can I see concrete examples of what generated tweets outperformed the topline? And what the actual topline tweets are? Again, I'm a little suspicious that the confounding variable here is something that is out of the author's control---and not the actual framing that the author has control over.
>
> **Regarding Timing**
>
> I am quite surprised that timing has a very small effect on persuasiveness! Time of day definitely has an impact on the number of users active on a platform---and even the content they engage with [1, 2]. Can you fit a simpler model here? Take a linear regression model, normalize the likes within a specific account, and map the time of day to the likes? Is the R^2 _really_ 0? This alone could be a big social science result (if true??) I'm happy to try to run this analysis myself, given the dataset.
>
> [1] Ozum et al. Tweets we like aren’t alike: Time of day affects engagement with vice and virtue tweets
>
> [2] Piccardi et al. Curious Rhythms: Temporal Regularities of Wikipedia Consumption
>
> **Regarding Clarity**
>
> Finally, I don't think adding _more_ results here really helped with clarity. I agree with EsZt here- I can't really recommend acceptance since this work was generally really hard for me to navigate personally.

---

> ### Comment · Reviewer_UfKL · 2024-11-25
> **Re: additional concerns and score update**
>
> I've gone ahead and re-read the paper, along with the reviews from the other authors. In light of additional concerns, I'm going to decrease my score.
>
> First, I am quite worried about the confounds in this work. I think there's plenty of prior work that argues for controls that this paper does not engage with. It's really hard to control for these exogenous factors---and I worry that the generative models are optimizing for things that are out of the writers control. I also agree with EsZt re: the conceptual framing; I really think this paper would be better positioned as an engagement prediction paper. This way, the authors can engage more with work in that area. I've cited two papers in my response that might point authors in the right direction---happy to cite more.
>
> Second, the oracle: why is the oracle trained on the test set too? Doesn't this reward overfitting to train? I mean, a model that does really well on the train set and then okay on the test set will be penalized more than a model that does just well on both (by the oracle). I also find it odd that for many of the human-eval tables, GPT-4o performs better than the fine-tuned models.
>
> Finally, clarity---this is also a major one: I think this paper could really use more focus! The results and discussion (Sec 5) lists many tables without engaging with what the results mean---readers are forced to go to the Appendix, and put together a narrative. I honestly think _removing_ some tasks, or even splitting up the benchmark and tasks into two different works, would help significantly.
>
> I want to end by saying this work is really cool! I think there's a lot of promise, but I'm leaning more towards a reject.

---

> ### Author Response · Authors · 2024-11-28
>
> > Regarding Timing: timing has a very small effect on persuasiveness! Time of day definitely has an impact on the number of users active on a platform. Can you fit a simpler model here? Take a linear regression model.
>
> 1. You are right, the timing of posts having a small effect on persuasiveness sounds counterintuitive. The Time of day does have an impact on the number of users active on a platform and the content they engage with [1,2]. We also show in the Chi square tests and Cramer's V analysis that there’s a significant impact of time of posting p=1.345 * 10^-5, but the effect is small in magnitude (<0.1).
> 1. The distinction between what we have done and what has been shown in literature comes from the various filters we have applied to construct the dataset. We illustrate the effect of each filter in the analysis below:
>
>
> We have applied the following filters (covered in Sections 2 and B.6):
> - **Business or Enterprise Accounts**: We collected 10k usernames from Google searches on Wikipedia entries that were marked as “business” or “enterprise”, we also identified 317 usernames that were found to be mislabeled as business or enterprise, based on Deberta Zero-Shot classification and manual filtering on top of it.
> - **<10 posts/day**: to filter accounts that do event based marketing, or whose tweets are only seen for shorter periods of time.
> - **Non-news, forecast, time dependent tweets**: We tag username descriptions from a few-shot (human-labeled) Deberta-v3 model, to filter out event based marketing channels, and accounts whose pairs showed significant difference (in content and likes) due to the timestamps. These reduce the number of tweets from 79M to 8.9M.
> - **Manual Filtering**: Finally, amongst the remaining 2457 accounts, we manually filtered out 112 accounts after clustering the tweets from each of these, resulting in 2345 accounts and 4M unique tweets
> - **Semantic Similarity**:  They should have high semantically similarity, the filters are given in Table 1. They should have a difference in likes of at least 20 percentile of the likes in their bimonthly period.
>
>
> To analyze the effect of each filter in diminishing the effect of time of posting in determining the engagement of the post, for each filter, for each username, we train a linear model on hour_of_day, day_of_week (X) and predict the monthly normalized likes (Y). We report the mean and standard deviation r^2 scores across all usernames.
>
>
> | Filter | Mean R^2 | Std R^2|
> |--------|------|-----|
> | Unfiltered | 0.11 | 0.005 |
> | Filter 1 (Enterprise/Business only accounts) | 0.11 | 0.005 |
> | Filters 1+2 (<10 posts a day) | 0.05 | 0.003 |
> | Filters 1+2+3 (Non-news, forecast, time dependent tweets) | 0.014 | 0.002 |
> | Filters 1+2+3+4 (Manual Filtering) | 0.014 | 0.002 |
> | Filters 1+2+3+4+5 = Final data | 0.013 | 0.001 |
>
>
> As seen from this table, the r^2 scores keep decreasing as we apply our filtering steps, thus showing that while in the original data, there is a significant and appreciable impact of time on engagement, in the filtered data, there is a very minor impact of time.
>
>
> To analyze the individual impact of each filter, we also perform this analysis in a different way where for each filter, we take all unfiltered samples and calculate the r^2 score for a classifier trained only on that data. Here are the results of that experiment:
> | Filter | Mean R^2 | Std  R^2|
> |--------|------|-----|
> | Unfiltered | 0.11 | 0.005 |
> | All except Filter-1 (Non-Enterprise/Business) | 0.17 | 0.015 |
> | All except Filter-2 (>=10 posts a day) | 0.14 | 0.002 |
> | All except Filter-3 (news, forecast, time dependent accounts) | 0.15 | 0.0003 |
> | All except Filter-4 (Manual Filtering) | 0.23 | 0.008 |
>
>
> For your ready perusal, we have attached the code and anonymous data to the supplementary. We have anonymized the tweet_ids, usernames as uids, and have given the timestamps as date_x and date_y for tweet with id_x and id_y. Tweets labeled as (Y=1) appeared on the right-hand side of a transsuasion sample (id_y > id_x).
> We have also attached a notebook where we run the binary classification using LogisticRegression.
> Thank you for engaging with us and asking us questions that can be scientifically answered. We would love to engage more with the reviewer to help clarify better.

---

> > ### Author Response · Authors · 2024-11-28
> >
> > > re: the conceptual framing; better positioned as an engagement prediction paper.
> >
> > Thank you for the suggestion, although we thought that this would be highlighted better as persuasion given the extent of generalisation across different domains, we have reconsidered it and will update this in the camera ready version of the paper.
> >
> > > Second, the oracle: why is the oracle trained on the test set too?
> >
> > We evaluate the strength of the oracle model through its performance on domain transfer experiments. A robust oracle should produce rankings on the PersuasionArena that generalize well across unseen persuasion benchmarks. Motivated by this, we conducted extensive experiments (Tables 7-12) across diverse human preference types, communication channels, KPIs, and content types. Our results show that the oracle model trained on both test and train datasets achieves the best performance. Specifically, the Kendall’s Tau ranking correlation between PersuasionArena and the unseen task benchmarks is 0.69, indicating strong generalization of the oracle’s persuasion rankings.
> >
> > For instance, as shown in Table 3, GPT-4o outperforms GPT-4 by 38 Elo points and GPT-3.5 by 160 Elo points on PersuasionArena. This corresponds to win rates of 55.45% and 71%, respectively, based on Bayes-Elo calculations. These findings align with the OpenAI GPT-o1 model card [2], which reports a 78.1% win rate on the ChangeMyView benchmark (compared to 71% on PersuasionArena). This consistency provides further external validation of the rankings produced by PersuasionArena’s oracle model, as the O1 model card serves as an independent source.
> >
> > To address concerns about potential overfitting caused by training the oracle on both test and train datasets, we also evaluated generative persuasion and content simulation directly from ground truth samples using NLG metrics such as BLEU-1, BLEU-2, ROUGE-1, ROUGE-L, and BERTScore. These metrics, which do not depend on the oracle, yielded rankings closely aligned with those in Tables 11 and 12. If the oracle were biased or overfitted, fine-tuned models would have demonstrated inferior performance. Instead, the observed consistency supports the robustness of our approach.
> >
> > Additionally, we performed an ablation study where the oracle was trained on the test set alone, which resulted in worse performance. Due to space constraints, we omitted these results from the paper but can include them in a future revision.
> >
> >
> > [2] o1 system card https://openai.com/index/openai-o1-system-card/

---

> > > ### Author Response · Authors · 2024-11-28
> > >
> > > > Regarding content confounds: First, even within the same brand or sub-category may often have tweets that focus on a specific feature. So for photoshop, my tweet might end up focus on two different features within Photoshop, confounding persuasiveness. Do the authors have an anonymous dataset link I can look through? I'd love to take a look at some samples myself.
> > > > about the confounds.
> > >
> > > To evaluate the semantic similarity (i.e. content confounders) of tweets in terms of specific features like products or tools, we have extracted the entities using spacy and measured the percentage of named entities that are consistent between input and output tweets, evaluated over 12k examples, both in the test dataset (87.1%) as well as generated tweets (92.1%) as shown in Table 15. We have also added an evaluation of consistency
> > > This shows that the generated tweets and the training data are both consistent in these terms. To further help you understand, we have added several samples for you below. For each example, Y was the transsuaded tweet.
> > >
> > > ---------------------
> > >
> > > Company:  nike
> > >
> > > **X:**  Boss the court with explosive speed. Lace up with the new #MercurialX, available now: <HYPERLINK> <HYPERLINK>
> > >
> > > **Y:**  Leave them searching for somewhere to hide. Discover the new #MercurialX, available now: <HYPERLINK> <HYPERLINK>
> > >
> > > ---------------------
> > >
> > > Company:  nokia
> > >
> > > **X:**  Better tools mean better outputs. Hello, #OZO. <HYPERLINK>
> > >
> > > **Y:**  Your toolbox matters. Meet #OZO, the tool for professional creators. <HYPERLINK>
> > >
> > > ---------------------
> > >
> > > Company:  sephora
> > >
> > > **X:**  Smoke and mirrors: Shop the new <USERNAME> Naked Smoky palette FIRST by signing up here: <HYPERLINK>
> > >
> > > **Y:**  Worth the wait: Be the first to shop the <USERNAME> Naked Smoky palette by signing up here: <HYPERLINK>
> > >
> > > ---------------------
> > >
> > > Company:  airbnb
> > >
> > > **X:**  Urgent: Help stop an unfair, anti-home sharing law in NY now <HYPERLINK> #IamAirbnb
> > >
> > > **Y:**  Unfair anti-home sharing law "sets a bullseye on middle-class New Yorkers." - <USERNAME>'s <USERNAME> <HYPERLINK> #IamAirbnb
> > >
> > > ---------------------
> > >
> > > Company:  asus
> > >
> > > **X:**  Everything creators and creative professionals need are right here in the #ASUS #ProArt StudioBook Series. Which one do you like best? #IFA19
> > >
> > > **Y:**  We know artists and creative professionals demand nothing but the best. We got you covered with the #ASUS #ProArt series. Time to power up your imagination! #IFA19
> > >
> > > ---------------------
> > >
> > > Company:  starbucks
> > >
> > > **X:**  Can you be-leaf Pumpkin Season is back at Starbucks?
> > >
> > >
> > > The one-and-only #PumpkinSpiceLatte is back for its 17th year alongside the #PumpkinCreamColdBrew and other fall favorites. Check out this link to learn more: <HYPERLINK>
> > >
> > > **Y:**  🎃 Welcome back, Pumpkin Spice Latte and Pumpkin Cream Cold Brew fans! Starting today all your Fall menu favorites have returned to stores. Here’s all the news you need to know:
> > >
> > > <HYPERLINK>
> > >
> > > ---------------------
> > >
> > >
> > > Company:  greenpeace
> > >
> > > **X:**  A reminder to <USERNAME>, <USERNAME> and other major plastic polluters:
> > >
> > >
> > > ⚠️ We need a global #PlasticsTreaty focusing on reducing the use of plastic ⚠️
> > >
> > >
> > > Otherwise the impacts of plastic on climate, environmental justice, and human health will not be solved.
> > >
> > >
> > > #BreakFreeFromPlastic <HYPERLINK>
> > >
> > > **Y:**  #BreakFreeFromPlastic
> > >
> > >
> > > The way that plastic waste is simply shipped off to countries in the global majority is quite colonial in its approach – dare I call it, #WasteColonialism.
> > >
> > >
> > > We need a strong #PlasticsTreaty to end the plastic industry's exploits🚫
> > >
> > >
> > > <HYPERLINK>
> > >
> > >
> > > ---------------------
> > >
> > > Company:  greenpeace
> > >
> > > **X:**  Over the next year we have the opportunity to create the largest protected area on Earth: #AntarcticSanturary <HYPERLINK>
> > >
> > >
> > > **Y:**  We have one year to create the largest ever protected area on Earth ... An #AntarcticSanturary <HYPERLINK>

---

> > > > ### Author Response · Authors · 2024-11-28
> > > >
> > > > > Can I see concrete examples of what generated tweets outperformed the topline? And what the actual topline tweets are?
> > > >
> > > > Sure, we have added the examples for the tweets, toplines and generated tweets below
> > > >
> > > > ---------------------
> > > >
> > > > **Company:**  clinique
> > > >
> > > > Original Tweet:  We’ve got a Sonic Brush & a full size Sonic Facial Soap up for grabs on our Facebook page! Enter here: <HYPERLINK>
> > > >
> > > > **Topline:** Win a Sonic Brush and full-size Sonic Facial Soap! Visit our Facebook page now to enter: <HYPERLINK>
> > > >
> > > > **Generated Tweet:**  ✨ Don't miss your chance to win a Sonic Brush and full-size Sonic Facial Soap! Head over to our Facebook page and enter now! 🌟 <HYPERLINK>
> > > >
> > > > ---------------------
> > > >
> > > > **Company:**  burberry
> > > >
> > > > Original Tweet:  The <USERNAME> Scarf Bar: Scottish-woven cashmere scarves in over 30 colours, personalised with your own initials <HYPERLINK>
> > > >
> > > > **Topline:** Discover The <USERNAME> Scarf Bar: luxurious Scottish-woven cashmere scarves in over 30 colors, personalized with your initials. Find yours today: <HYPERLINK>
> > > >
> > > > **Generated Tweet:**  Elevate your style with The <USERNAME> Scarf Bar! 🌟 Wrap yourself in luxury with our Scottish-woven cashmere scarves, available in over 30 stunning colors. Each piece can be personalized with your initials, making it uniquely yours. Discover your next statement piece here: <HYPERLINK>
> > > >
> > > > ---------------------
> > > >
> > > >
> > > >
> > > > **Company:**  greenpeace
> > > >
> > > > Original Tweet:  Plastic bags was a great start, but New Zealand needs a comprehensive national strategy to eliminate throwaway single-use plastic.
> > > >
> > > >
> > > > Sign and share: <HYPERLINK>
> > > >
> > > >
> > > > #BreakFreeFromPlastic
> > > >
> > > >
> > > > **Topline:** Banning plastic bags was just the start. New Zealand needs a national strategy. Sign and share: <HYPERLINK>
> > > > #BreakFreeFromPlastic
> > > >
> > > >
> > > >
> > > > **Generated Tweet:**  Plastic bags were just the beginning! New Zealand urgently needs a comprehensive national strategy to eliminate all single-use plastics. Let's make a real change!
> > > >
> > > >
> > > > Sign and share: <HYPERLINK>
> > > >
> > > >
> > > > #BreakFreeFromPlastic
> > > >
> > > > ---------------------
> > > >
> > > > **Company:**  louis vuitton
> > > >
> > > > Original Tweet:  Watch tomorrow’s #LouisVuitton Women’s Spring-Summer 2019 Fashion Show by <USERNAME> live on Twitter starting at 8pm Paris time. Set a reminder now.
> > > > #LVSS19
> > > > <HYPERLINK>
> > > >
> > > > **Topline:** Tune in tomorrow at 8 PM Paris time to watch the #LouisVuitton Women’s Spring-Summer 2019 Fashion Show live on Twitter. Don’t miss it!
> > > > #LVSS19 <HYPERLINK>
> > > >
> > > > **Generated Tweet:**  🌟 Don’t miss the dazzling #LouisVuitton Women’s Spring-Summer 2019 Fashion Show, featuring exclusive looks by <USERNAME>! Join us live on Twitter tomorrow at 8 PM Paris time. ✨ Set your reminders and be part of the fashion magic!  #LVSS19 [HYPERLINK]
> > > >
> > > > ---------------------
> > > >
> > > > **Company:**  nike
> > > >
> > > > Original Tweet:  Boss the court with explosive speed.
> > > >
> > > >
> > > > Lace up with the new #MercurialX, available now: <HYPERLINK> <HYPERLINK>
> > > >
> > > >
> > > > **Topline:** Leave them searching for somewhere to hide. Discover the new #MercurialX, available now: <HYPERLINK> <HYPERLINK>
> > > >
> > > > **Generated Tweet:**  Unleash your game with lightning speed! ⚡️
> > > >
> > > >
> > > > The all-new #MercurialX is here, and they're ready to elevate your performance on the court! Don't miss out—grab your pair now: <HYPERLINK>
> > > >
> > > > ---------------------
> > > >
> > > > **Company:**  louis vuitton
> > > >
> > > > Original Tweet:  The #LouisVuitton #LVFW18 Fashion Show will be live from #PFW next Tuesday. Stay tuned!
> > > > <HYPERLINK>
> > > >
> > > > **Topline:** The #LouisVuitton #LVFW18 Fashion Show goes live from #PFW next Tuesday. Stay tuned for all the fashion highlights!
> > > > <HYPERLINK>
> > > >
> > > > **Generated Tweet:**  🌟 Get ready for the #LouisVuitton #LVFW18 Fashion Show, live from #PFW next Tuesday! Stay tuned! #FashionForward #ParisFashionWeek
> > > >
> > > >
> > > > <HYPERLINK>
> > > >
> > > > ---------------------
> > > >
> > > > **Company:** clinique
> > > >
> > > > Original Tweet: Black Honey — or black magic? ✨Maybe both. Reply with why you love this confoundingly-flattering TikTok fav, then get yours now by purchasing on <HYPERLINK> before everyone else.#Clinique #BlackHoney #CliniqueBlackHoney #TikTokmademebuyit #tiktokproducts #viralmakeup
> > > >
> > > >
> > > > **Topline:** Black Honey—TikTok’s viral sensation. Tell us why you love it, and grab yours before it’s gone: <HYPERLINK>
> > > > #Clinique #BlackHoney #TikTokMadeMeBuyIt
> > > >
> > > > **Generated Tweet:** Black Honey — magic in a tube? ✨ Maybe! Share why this TikTok-favorite lipstick has you obsessed, and grab yours at <HYPERLINK> before it sells out. 💋 #Clinique #BlackHoney #CliniqueBlackHoney #TikTokMadeMeBuyIt #ViralMakeup

---

> > > > > ### Author Response · Authors · 2024-11-28
> > > > > **Topline Examples Continued**
> > > > >
> > > > > **Company:**  greenpeace
> > > > >
> > > > > Original Tweet:  Forest protection is climate protection!
> > > > >
> > > > >
> > > > > #DanniBleibt We stand with you. We cannot afford to destroy a healthy forest for another highway.
> > > > >
> > > > >
> > > > > Join the mobilization >> <HYPERLINK> <HYPERLINK>
> > > > >
> > > > > **Topline:** Healthy forests are essential for a healthy climate. Stand with #DanniBleibt—don’t let another highway destroy ecosystems. Join us: <HYPERLINK> <HYPERLINK>
> > > > >
> > > > >
> > > > >
> > > > >
> > > > > **Generated Tweet:**   Protecting forests means protecting our climate! We stand with #DanniBleibt—let’s not sacrifice a thriving ecosystem for another highway.
> > > > >
> > > > >
> > > > > Join the movement to safeguard our future!  <HYPERLINK> <HYPERLINK>

---

> > > > > > ### Author Response · Authors · 2024-11-30
> > > > > > **Anticipating Your Participation as Reviewer-Author Discussion Deadline Approaching**
> > > > > >
> > > > > > We express our sincere gratitude to the reviewer for dedicating time to review our paper. We have provided comprehensive responses to all the concerns. As the discussion deadline looms within 3 days, we would like to inquire if our responses have adequately addressed your questions. We are more than willing to address any concerns and ensure a comprehensive resolution. Thank you for your time and consideration.

---

> > > > > > > ### Author Response · Authors · 2024-12-03
> > > > > > > **Anticipating Your Participation as Reviewer-Author Discussion Deadline Approaching**
> > > > > > >
> > > > > > > We express our sincere gratitude to the reviewer for dedicating time to review our paper. We have provided comprehensive responses to all the concerns. As the discussion deadline looms within 1 day, we would like to inquire if our responses have adequately addressed your questions.
> > > > > > >
> > > > > > > We have also attached the code and dataset you had asked us to share and are more than willing to address any concerns and ensure a comprehensive resolution.
> > > > > > >
> > > > > > > Thank you for your time and consideration.

---

### Meta-Review · Area_Chair_3RVS · 2024-12-20

**Metareview:**

**Summary:**

The authors study one-dimensional transformation of persuasiveness in natural language communication, or transsuasion, to control for semantics and context (e.g. time/receiver) of communication while adjusting persuasiveness. They contribute two new datasets for studying persuasion, PersuasionBench and PersuasionArena, which were collected by searching for similar marketing content spread on enterprise social media accounts. They evaluate LLM persuasiveness in (1) a generative setting where they observe if a LLM can effectively transform content to make it more persuasive and (2) a simulative setting to see if LLMs understand factors contributing to low or high like counts. They find that LLMs lag in simulative capabilities with barely above chance accuracy, but perform well at generating persuasive content when compared against human marketers.

**Strengths:**

- The work demonstrates interesting and potentially impactful results that LLM persuasiveness can be improved through targeted instruction tuning rather than being primarily influenced by model scale.

- Comprehensive evaluation with a range of LLMs.

- The resources provided seem like they would be beneficial to the broader research community and industry partners evaluating online persuasion tactics and LLM-based advertising.

**Weaknesses:**

- The organization of the paper and in particular the use of "simulation" to describe experiments is confusing.

- Some reviewers expressed concern over the paper's focus on indirect measures of persuasion through social media engagement prediction, rather than directly measuring persuasion from the psychology literature referenced.

**Additional Comments On Reviewer Discussion:**

Most of the reviewers are leaning towards acceptance. I agree with reviewers UfKL and EsZt that the paper's writing suffers from a lack of clear organization around the tasks covered. It is challenging to discern the distinction between the generative and simulation results (at first it is easy to assume simulation would also be predictive), and I would encourage the authors to put more effort into polishing the paper and potentially renaming "generative" vs. "simulation" experiments before publication.

The issue of time of day being a confounding factor (raised by reviewer UfKL) has been comprehensively resolved in the authors' rebuttal, and it seems reasonable the effect would be relatively small given that the data came from companies' marketing content (presumably they would already have some prior about advantageous times to post). It should be addressed as a potential limitation of the work, and an aspect to focus more on in future work, but I don't see this as a major issue that would hinder publication of this paper.

---

### Decision · Program_Chairs · 2025-01-22

Accept (Poster)